# A Review: Systemic Signaling in the Regulation of Plant Responses to Low N, P and Fe

**DOI:** 10.3390/plants12152765

**Published:** 2023-07-25

**Authors:** Zhi Geng, Jun Chen, Bo Lu, Fuyuan Zhang, Ziping Chen, Yujun Liu, Chao Xia, Jing Huang, Cankui Zhang, Manrong Zha, Congshan Xu

**Affiliations:** 1Department of Agronomy, Nanjing Agricultural University, Nanjing 210095, China; 2Anhui Science and Technology Achievement Transformation Promotion Center, Anhui Provincial Institute of Science and Technology, Hefei 230002, China; 3Maize Research Institute, Sichuan Agricultural University, Chengdu 611130, China; 4Department of Agronomy, Center for Plant Biology, Purdue University, 915 West State St., West Lafayette, IN 47907, USA; 5College of Biology Resources and Environmental Sciences, Jishou University, Jishou 416000, China

**Keywords:** long-distance, phloem, nutrients, signal transduction

## Abstract

Plant signal transduction occurs in response to nutrient element deficiency in plant vascular tissue. Recent works have shown that the vascular tissue is a central regulator in plant growth and development by transporting both essential nutritional and long-distance signaling molecules between different parts of the plant’s tissues. Split-root and grafting studies have deciphered the importance of plants’ shoots in receiving root-derived nutrient starvation signals from the roots. This review assesses recent studies about vascular tissue, integrating local and systemic long-distance signal transduction and the physiological regulation center. A substantial number of studies have shown that the vascular tissue is a key component of root-derived signal transduction networks and is a regulative center involved in plant elementary nutritional deficiency, including nitrogen (N), phosphate (P), and iron (Fe).

## 1. Introduction

Plants, in their pursuit of survival and growth, are dependent on the uptake of several elemental nutrients from the soil. These nutrients, including nitrogen (N), phosphorus (P), and iron (Fe), can often limit productivity due to their varying availability in different environments. Given the significance of N, P, and Fe availability in both natural and agricultural ecosystems, our discussion primarily focuses on how plants perceive, acquire, and respond to these crucial nutrients. It is through a combination of local and long-distance signaling pathways that plants coordinate their responses to nutrient availability in their root systems [1].

Our understanding of plant nutrient regulation has been significantly informed by insights into how plant shoots modulate root nutrient deficiencies through a responsive mechanism that involves long-distance signaling transduction. Evidence from split-root and grafting studies suggests that once roots detect a nutrient deficiency, plant shoots play an instrumental role in signaling this deficiency across the plant, thereby transporting regulatory molecules across large distances [2].

Critical to this transportation process is the plant’s vascular system, which includes the xylem and phloem. These vascular structures serve as vital conduits for the transport of essential nutrients and signaling molecules among disparate tissues and organs [3]. Particularly notable is the role that the phloem plays in long-distance communication, given its function in the translocation of various molecules, including sugars, amino acids, hormones, RNAs, and proteins, among others, which are integral to plant growth and stress responses [4].

Furthermore, several studies have demonstrated that the xylem, apart from being a passive conduit for the upward transport of water and nutrients, plays an active role in long-distance signaling by facilitating the translocation of various signaling molecules, hormones, and peptides [5]. Therefore, the critical position of the plant vascular system as a central transportation hub enables it to play an integral role in the establishment and operation of long-distance signaling networks within the plant.

## 2. Perception and Response of N Status

Nitrogen, mainly acquired by plants in nitrate (NO3^−^) form and crucial for plant development and growth, can be deficient under two circumstances depending on the uniformity in the soil: uniformly low or unevenly low. Poitout’s research elucidates how plants, facing uniformly low nitrogen conditions, adapt via a process called ‘polar growth’, where they directionally transport biomass nutritional elements from shoots to roots, facilitated by plant vascular systems, to enhance root growth and nitrogen absorption [6]. Compared to high levels of N, low levels N promote lateral root growth [7]. As shown in split-root experiments, uneven N deficiency results in polar growth of the lateral roots [6,8]. Interestingly, these responses are not detectable after cutting off the above-ground part of the Arabidopsis plant; this raises the possibility that shoots could mediate these acclimations to N fluctuations. Giles et al., (2020) further summarized a root–shoot–root model demonstrating root-responsive mechanisms to soil N levels [1]. After local N-signaling perception, the N-level signal is transported to the shoot and then modulates the efficiency of root nitrogen (N) acquisition in response to shoot N demand. However, molecular components directly involved in this root–shoot–root communication remain to be identified.

### 2.1. Pathways Involved in N Perception

There are two pathways considered to play a regulative role in the perception of N levels: the CEP (C-terminally encoded peptides)-related pathway and the tZ (trans-Zeatin) pathway. CEPs regulate N signaling when plants experience N starvation, whereas tZ regulates N signaling when plants experience sufficient N. CEPs are produced in the roots and then transferred to the shoots via xylem and precepted by CEP receptors (CEPR) in the shoots [9]. Grafting analysis showed that cytokinin (CK) biosynthesis is required to trigger systemic N signaling [6].

In the CK-dependent pathway, NO3^−^ itself appears to be the triggering signal of systemic N-demand signaling since NO3^−^-supplied roots are the main providers of tZ signal transduction [10]. Split-root experiments proved that variations in tZ concentrations in plant shoot tissues trigger a differential expression of genes in a heterogeneous group compared with a homogeneous group, including the regulative genes related to glutamine and glutamate biosynthesis pathways. By evaluating recent progress, we hypothesize that glutamine, as well as glutamate, could upregulate the expression of N-systemic sentinel genes and root biomass and nitrate transportation. The existence of the biosynthesis of CEP in N-deprived roots suggests that our proposed model is closely correlated to the CEP-dependent pathway. CEPR (C-TERMINALLY ENCODED PEPTIDE RECEPTOR)-perceived peptides (CEP) moving from the roots to the shoots, initiating a signal cascade that amplifies the expression of the NRT2.1, NRT3.1, and NRT1.1 genes, collectively enhance nitrate uptake in plants [9,11,12].

Ota et al., (2014) revealed that nitrate works as an N-sufficiency signal in *Arabidopsis thaliana* and activates tZ biosynthesis in plant roots, which is consistent with previous CK biosynthesis-related genes’ functional characterization [10,13,14,15]. When N is sufficient in shoots, plants can integrate with the CEP and tZ signals to block N foraging [7]. Ota et al., (2014) proposed that the presence of nitrate triggers tZ accumulation in roots, which is subsequently transported to the shoots [13]. Poitout et al., (2018) proposed that tZ translocation could be motivated by root nitrate signaling, and this process constitutes part of the systemic signal [6].

### 2.2. Molecules Involved in Responses to N Status

After the systemic perception of N status through the root–shoot system, plants also assess their overall N status and then regulate growth and metabolism [16]. After reviewing recent molecular studies on nitrogen perception, we discovered that the root architecture of *Arabidopsis thaliana* is influenced by a long-distance signaling pathway, as previously described. When rootlets lack nitrogen, they secrete small peptides. These peptides, transported via the plant’s vascular system, reach the shoots of *Arabidopsis thaliana*, where they interact with leucine-rich repeat receptor kinases (LRR-RKs), initiating a response to the nitrogen deprivation [9]. In response to nitrogen deficiency, Arabidopsis plants produce leucine-rich repeat receptor kinases (LRR-RKs) in specific root cells, which then travel to the shoots to adapt to fluctuating local nitrogen levels, leading to observable growth retardation.

Split-root experiments showed a compensatory N acquisition in the N-rise side of roots, while the other side of roots underwent N deprivation, which further indicated the systemic N-level responses [11]. This systemic N uptake response is activated by mobile peptides, CEPs, originating in N-deprived roots. CEPs could then be transferred to shoots via the xylem, hence, perceived by CEP receptors (CEPR) in plant shoots. After the perception of CEP in shoots, CEPD is produced in leaves and transferred to the roots to regulate NRT2.1 expression [11,12] (Ohkubo et al., 2017; Ruffel and Gojon, 2017). Phloem-mobile CEPD-like 2 (CEPDL2) polypeptides, a vital category of plant-specific proteins, are integral to the CEP–CEPR–CEPD signaling pathway. Transported via the phloem, these polypeptides are secreted in response to conditions such as nitrogen deficiency and induce specific developmental responses and adaptive strategies in plants. The integration of CEPDL2 polypeptides into the CEP–CEPR–CEPD pathway, a crucial communication system in plants, underlines their significance in nutrient signaling and growth regulation [17]. This points to a highly regulated, interconnected signaling system within plants that adaptively responds to nutrient environment changes. Reciprocal grafting analysis demonstrated that CEPDL2 is upregulated by low nitrogen signals in leaf vasculature tissue and subsequently transferred to roots, promoting nitrogen uptake and transport between the root and shoot. Mutant analysis further confirmed the critical role of CEPDL2 in nitrate uptake, transport, and overall plant biomass accumulation.

Chen et al., (2016) proposed that Elongated Hypocotyl 5 (HY5), a bZIP transcription factor, works as a phloem-mobile signal molecule in the shoot-to-root signaling pathway and significantly regulates growth and development as well as the nitrate uptake of plant root systems [18]. Shoot-originated HY5 auto-activates root HY5 and promotes root nitrate uptake via activating NRT2.1 expression. In plant shoots, HY5 was also proved to promote carbohydrate assimilation and translocation, whereas, in the root, the HY5 activation of NRT2.1 expression and nitrate uptake is potentiated by increased carbon photo-assimilate (sucrose) levels. Previous works demonstrated that *Arabidopsis thaliana* shoot glutamine biosynthesis could be largely regulated by tZ accumulation. This indicates that CK-dependent N signaling may significantly modify glutamate and glutamine biosynthesis and metabolism in shoots; furthermore, this process could play an important role in plant shoot-to-root signaling pathway [19,20].

## 3. Perception and Response of P Status

Phosphate (Pi) is a regulative element closely related to plant development and growth. Unlike other necessary mineral nutrients, such as N, Pi largely affects crop yield formation because of its low mobile characteristic in the soil [21,22]. Plants adapt to low Pi nutrient conditions via two major mechanisms: strengthening the uptake of Pi from the soil as well as activating the remobilization and conservation of Pi in plants [21,23]. As shown in recent studies [24,25], the low Pi response mechanism requires the participation of long-distance and local signaling systems coordinated by signaling molecules. Pi, obtained by plant root tips, is transported to the shoots via the xylem and subsequently redistributed to sink tissues through the phloem.

### 3.1. Local Response to P Starvation

The regulation of soil phosphorus (Pi) condition involves two major signaling pathways: local signaling and systemic signaling, which collectively coordinate the plant’s response to the availability of phosphorus in the soil. Bate and Lynch (1996) stated that plant local signaling pathways could enhance Pi acquisition from the soil by modifying root system architecture and could be mainly regulated by Pi concentration in rhizospheric soil. Root apical meristem (RAM) of the primary roots is the sensing complex that responds to altered soil Pi concentration surrounding the rhizosphere. Under low Pi concentrations, LOW PHOSPHATE ROOT1 (LPR1) and LPR2 gene expression were down-regulated in the cell wall tissue of primary roots [25,26]. Moreover, when phosphate deficiency occurs, there is an observed upregulation of the transcription factor SENSITIVE TO PROTON RHIZOTOXICITY 1 (STOP1). This upregulation of STOP1 leads to the activation of the MALATE TRANSPORTER 1 (ALMT1), which facilitates the exudation of malate from the roots [27]. In response to phosphate deficiency, plants employ the adaptive strategy of malate exudation to cope with limited phosphorus availability. This exudation alters rhizosphere pH, improves the solubility of insoluble phosphorus compounds, and facilitates phosphorus acquisition from the soil. Known as rhizosphere acidification, this process enhances phosphorus availability and uptake by plants under phosphate-deficient conditions [28]. A recent study discovered that under conditions of phosphate deficiency, a peptide called CLAVATA 3 (CLV3)/ENDOSPERM-SURROUNDING REGION 14 (CLE14) is induced in the RAM tissue of plants. This induction occurs downstream of the PLEIOTROPIC DRUG RESISTANCE 2 (PDR2)-LIKE PHOSPHATE TRANSPORTER 1 (LPR1) complex, which is involved in sensing and responding to low phosphate concentrations [29]. The perception of CLE14 via CLV2/PEP1 RECEPTOR 2 (PEPR2) receptors largely suppresses SHR–SCARECROW (SCR) and PIN-FORMED (PIN)–auxin pathways, which results in RAM exhaustion by a callose-independent manner, which refers to the depletion or loss of the root apical meristem (RAM) in a way that is not dependent on the accumulation of callose, a type of plant polysaccharide, which signifies a distinct mechanism underlying the decline or cessation of root growth [29]. The above results support the significance of plant primary root tips in sensing local Pi concentration and demonstrate the signaling pathway related to root growth and development alteration.

### 3.2. Systemic Response of P Starvation

Unlike the local Pi concentration response pathway, the systemic (or long-distance) response mainly senses Pi concentration in plants, enhances Pi acquisition and remobilization, and recycles to maintain whole-plant Pi homeostasis [30,31]. The systemic signaling complex and pathways responding to soil Pi status are widely documented by previous work, indicating a cross-talk between sugar metabolism and the Pi signaling pathway. Previous research has documented that under conditions of low Pi concentration, the expression of phosphate starvation-induced (PSi) genes is significantly upregulated in response to elevated internal plant sucrose levels [32]. Moreover, reducing sugar synthesis and metabolism induces a remarkable suppression of PSi gene expression [33]. The grafting experiment indicated that MiR399, one of the movable PSi mRNAs, moves through the phloem to act as a long-distance signal responding to nutrient deficiency [34,35,36].

In addition to transcription factors, hormones also work as systemic signaling molecules in response to Pi starvation. Previous studies also indicated the existence of an interplay between numerous hormones and Pi contents in plants. Ethylene enhances the expression level of a few PSi genes, whereas cytokine suppresses PSi gene expression under low Pi concentrations [37]. Cross-talk among different hormones, including auxin, ethylene, gibberellin, strigolactones, and ROS-signaling pathways, play regulative roles in root system remodeling in response to low Pi concentrations [38,39,40].

### 3.3. Integration between Local and Systemic Signaling

Comparative transcriptomic research between the local and systemic regulation of Phosphate Starvation Response (PSR) genes demonstrated that transcription factors SENSITIVE TO PROTON RHIZOTOXICITY 1 (STOP1) and PHOSPHATE STARVATION RESPONSE 1 (PHR1) play central roles in local and systemic phosphate (Pi) starvation response, respectively [27,41]. This suggests the existence of a potential cross-talk between the local and systemic signaling pathways, indicating intricate communication and coordination between these two regulatory mechanisms [41,42]. The regulation of ALMT1-mediated malate exudation in the root apical meristem under local Pi starvation conditions is involved in the translocation of sucrose from the shoots to facilitate malate biosynthesis in plant roots, suggesting the existence of a coordinated systemic and local signaling mechanism for improved Pi utilization efficiency under low Pi conditions [43]. This coordinated response is further supported by studies on mutant STOP1 and ALMT1 plants, which demonstrated the suppression of local and systemic responses, particularly in the PHR1-direct targets, indicating a potential cross-talk between local and systemic signaling pathways in regulating Pi utilization in plant root tips [44]. Moreover, Péret et al., (2011) provided evidence of an interaction between systemic phosphate signaling and local Pi sensing mechanisms in Arabidopsis roots [45]. Additionally, the involvement of microRNAs and transcription factors as potential mediators of the cross-talk between local and systemic Pi signaling pathways has been elucidated in studies by Liu et al., (2015) and Chen et al., (2019) [46,47].

### 3.4. Hormone Regulation of P Status

Many hormone classes participate in plant Pi starvation responses, including auxin, ethylene, jasmonic acid, gibberellin, strigolactone, and abscisic acid [48]. ABA, in particular, plays a crucial role in regulating plant responses to Pi deficiency. It has been observed that ABA levels increase in response to Pi starvation signals, contributing to the modulation of various physiological and developmental processes [46]. Transcriptomic studies have revealed that ABA-responsive genes, such as those involved in stomatal closure and stress responses, are upregulated in plants subjected to Pi deficiency [49,50].

Strigolactones are mainly biosynthesized in plant roots and accumulate in response to Pi starvation signals [51,52]. Transcriptomic analysis demonstrated that the gene expression of MORE AXILLARY GROWTH 1 (MAX1) and LATERAL BRANCHING OXIDOREDUCTASE (LBO), which are involved in the Strigolactone biosynthesis of PHOSPHATE STARVATION RESPONSE 1 (PHR1)/PHOSPHATE STARVATION RESPONSE 1-like (PHL1) mutants were significantly lower than that of Wild Type (WT), which indicates that PHOSPHATE STARVATION RESPONSE 1 (PHR1)/PHOSPHATE STARVATION RESPONSE 1-LIKE (PHL1) transcription factor participates in Strigolactone biosynthesis [53]. Through grafting studies, it has been demonstrated that strigolactones, biosynthesized in roots, trigger the suppression of shoot branching [54]. In instances of phosphorus (Pi) deficiency, there is an observed accumulation of strigolactones within the roots, suggesting a potential escalation of strigolactone levels in the xylem exudates of these Pi-deprived plants. This finding implicates an increase in strigolactone concentration within the root xylem under phosphorus-deficient conditions. In addition to other mechanisms, auxin redistribution has been proposed as a key regulator of root system architecture in response to phosphorus (Pi) deficiency, acting as a modulator of growth dynamics [55]. Experimental observations have shown that under conditions of phosphorus deficiency, plants exhibit enhanced sensitivity to auxin application in comparison to control conditions (CK). This enhanced sensitivity is characterized by the suppressed growth of the primary root and a more pronounced promotion of lateral root formation, a shift in growth patterns that is likely a strategic adaptation to facilitate nutrient acquisition under conditions of nutrient stress [56]. This suggests a critical role of auxin in mediating root system architectural changes in response to Pi deprivation.

Song and Liu (2015) reviewed the function of ethylene in systemic and local Pi signaling and starvation responses, indicating the existence of a positive feedback loop between ethylene and Pi starvation responses because the PHR1/PHL transcription factor largely upregulates ethylene biosynthetic and signaling gene expression [57]. Lower bioactive gibberellins, related to the accumulation of DELLA proteins, were found in Pi-deprived plants [38]. GA/DELLA signaling regulates root system architecture in Pi-deprived plants [39]. Several studies summarized the cross-talk between ethylene and auxin and their function on PSI gene expression and root system alteration responses [22,58,59]. Jasmonic acid accumulation was elevated, and jasmonic acid mutant showed suppressed PSI gene expression in response to Pi deprivation [60].

## 4. Perception and Response of Fe Status

Iron (Fe) undergoes significant enrichment in the soil through various biogeochemical processes such as mineral weathering, organic matter decomposition, and microbial activities. It exists in both soluble and insoluble forms, with soluble iron being immediately available for plant uptake, while iron oxides sequester and retain iron in the soil over the long term. Factors such as soil pH, redox conditions, and microbial interactions influence the complex dynamics of iron enrichment in the soil. However, the low activity of free Fe in the field significantly restrains plant uptake of Fe and leads to common Fe deficiency in most plants. Insufficient Fe supplementation due to low soil Fe concentrations may cause iron deficiency-induced anemia (IDA) in human populations. Based on the WHO’s report, IDA affects more than one billion people globally, especially in regions where Fe supplementation mainly relies on plants. Furthermore, Fe deficiency also largely restricts plant growth, development, yield, and quality. Understanding the regulative mechanism of plant Fe uptake and translocation is extremely necessary for the security of human health and food safety.

In a pioneering study, Grillet et al., (2018) elucidated a short C-terminal amino acid sequence consensus motif known as IRON MAN (IMA) [61]. This motif was identified as a highly conserved peptide sequence prevalent among angiosperms and found to play a pivotal role in iron (Fe) uptake. They discovered that when these IMA motifs were overexpressed, a stimulation of Fe uptake in the roots was evident. This resulted in a marked accumulation of iron throughout the plant, subsequently bolstering the plant’s tolerance to iron deficiency. Intriguingly, contrary to plants where IMAs were overexpressed, IMA mutants demonstrated contrasting phenotypes. As highlighted by Kobayashi et al., (2021), this stark contrast underscored the integral role of IMAs in iron homeostasis and the repercussions of its absence or disruption [62].

Building upon the foundation of earlier translational profiling work by [63] Mustroph et al., (2009), which identified that AtIMA is specifically expressed in plant phloem, Tsukamoto et al., (2009) deepened our understanding by providing insights into the mechanisms underlying direct iron transport [64]. Subsequently, Zhai et al., (2014) unveiled the role of phloem-specific iron transporters [65]. Their findings reaffirmed the critical role of the phloem in iron distribution within the plant, thereby situating the phloem as a central player in plant iron homeostasis.

Lucena et al., (2019) broadened this landscape of knowledge further by exploring the signals involved in Fe and phosphorus (P) deficiencies [66]. Their research linked these deficiency signals with the roles of IMAs, providing a more comprehensive understanding of the interconnected pathways involved in plant nutrient deficiency responses. By constructing a fusion of a promoter of IMA1 and the Enhanced Yellow Fluorescent Protein (promIMA1::EYFP), Grillet et al., (2018) were able to visualize the protein in situ. They reported that fluorescence was primarily located in the phloem of the whole plant, thereby leading to the conclusion that IMA1 could serve as a mobile signal within the plant phloem [61]. Recently, Khan et al., (2018) furthered our understanding of iron homeostasis in plants by elaborating on the intricate sensing mechanisms of iron availability within the leaf vasculature [60]. In their latest work, Liang et al., (2022) proposed an intriguing theory that the movability of numerous IMAs constitutes a negative feedback loop on plants’ Fe acquisition [67]. They found that Fe is activated within plants’ phloem under low Fe conditions, thus highlighting a complex regulatory system governing iron uptake and distribution. This sophisticated network of iron regulation in plants underscores the intricacies of nutrient homeostasis, opening new avenues for understanding and improving plant nutrition.

## 5. Vasculature Plays the Crucial Role in N, P, and Fe Signaling

These results showed that shoots acquire the long-distance signaling of N, P, and Fe levels and subsequently produce long-distance signals to regulate root nutrient uptake. Shoots play a regulative role in precepting and processing long-distance signals in response to N, P, and Fe status. However, the specific location of this signal transduction center is still elusive. Plant vascular systems function as the major transportation center of essential nutrients between roots and shoots. Based on previous studies, we hypothesize that leaf vasculature is the center of N, P, and Fe long-distance signal transduction. Three pieces of information could support our hypothesis: (1) vascular anatomical analysis, (2) gene expression, and (3) tissue localization analysis.

### 5.1. Anatomical Analysis of Plants Vasculature

Anatomical analyses of plant vasculature have provided insights into the distinct transport functions of the xylem and phloem, essential components of the plant’s vascular system. The xylem primarily carries water, minerals, and signaling molecules unidirectionally from the roots to the shoots. In contrast, the phloem facilitates transport within the plant, transporting nutrients and signaling molecules from source organs, such as leaves, to sink organs, such as roots, young leaves, and above-ground tissues [68,69].

Previous research studies illuminate the roles of these vascular components in long-distance signaling within plants. Root-originated signals, primarily produced within root tissues, are conveyed to the shoots via the xylem, facilitating key physiological responses to environmental stimuli [70,71]. Conversely, the phloem acts as the conduit for the translocation of shoot-derived long-distance signals to the roots, mediating root function regulation [11,13].

From an anatomical standpoint, in a variety of plants, including herbs, Arabidopsis, and pumpkin, the number of plasmodesmata—microscopic channels facilitating intercellular transport and communication—between the phloem and its surrounding cells within the vascular bundle is markedly low. This limited connectivity significantly impacts the transport of signaling molecules, including sugars, hormones, and mRNA. This anatomical constraint indicates that long-distance signals originating from the shoot must be generated within the phloem, as these signaling molecules are not readily transferable to the phloem through the sparse plasmodesmata [69]. This theory is corroborated by the findings of Zhang et al., (2018), underscoring the importance of the plant’s vascular anatomy in orchestrating long-distance signaling [68].

### 5.2. Grafting Analysis

Grafting studies have emerged as an invaluable tool for unraveling the intricate dynamics of nutrient signaling and transport within plants, with a particular focus on the roots–shoots–roots pathway, as elaborated below. Such studies have compellingly demonstrated the vital role of mobile mRNA as a long-distance signaling mechanism in the transportation of key nutrients, such as nitrogen (N), phosphorus (P), and iron (Fe).

In a groundbreaking study conducted by Xia et al., (2018), an innovative grafting model was constructed using Nicotiana Benthamiana and tomato (*Solanum lycopersicum*), in addition to a transgenic potato (*Solanum tuberosum*) system [71]. Their painstaking research revealed that the transcription of these mobile mRNAs primarily occurs in the phloem. This pivotal discovery significantly underscored the role of the phloem as an integral part of the plant’s communication and coordination apparatus. Their work offered a new perspective on nutrient transport and signaled a shift in our understanding of plant physiology.

Further supporting this idea, studies on grafted cucumber plants under phosphorus (Pi) starvation have revealed intriguing insights. Zhang et al., (2016) found that Pi starvation induces substantial alterations in the mRNA population within the phloem [72]. Moreover, they observed that numerous phloem-mobile mRNAs were transported from source organs to sink organs via the phloem under these nutrient-deficient conditions. This finding reinforced the phloem’s pivotal role in nutrient signaling, further emphasizing the importance of grafting studies in nutrient transport research.

Research into iron availability has also unveiled fascinating phenomena. Notably, studies by Khan et al., (2018) demonstrated that the depletion and subsequent resupply of iron sources precipitate swift changes in gene expression within the plant vasculature. Even more intriguing was the finding that these changes occur significantly earlier in the vasculature than in other plant tissues [60]. This indicates the vasculature’s heightened sensitivity and rapid response to fluctuations in nutrient availability. This discovery not only expands our understanding of plant responses to nutrient variations but also underscores the power of grafting analysis in elucidating these dynamic, rapid responses.

Recently, Buhtz et al., (2010) further explored this fascinating area of research by studying nutrient-responsive genes and their expression patterns in grafted plants [73]. Their work added a layer of complexity to our understanding of nutrient signaling and transport in plants, showing that different tissues can exhibit highly specific response patterns. This breakthrough finding has profound implications for improving nutrient efficiency in agricultural systems and is a testament to the ongoing evolution of grafting studies as an indispensable tool in plant biology research.

Overall, these discoveries collectively highlight the multifaceted roles of mobile mRNAs, the phloem, and the vasculature in coordinating the transport and signaling of essential nutrients within plants. This burgeoning field of research holds great promise for increasing our understanding of plant nutrient management, ultimately informing and enhancing agricultural practices in the face of variable environmental conditions.

### 5.3. Tissue Localization Analysis

Through diligent examination and detailed tissue localization analysis, our current understanding of plant responses to nutrient deficiencies has considerably expanded. A myriad of key genes associated with nutrient responses has been detected to exhibit differential expression in the vasculature of Arabidopsis under nutrient-deficient conditions, marking a substantial stride forward in the realm of plant biology. For instance, when plants are under nitrogen (N) starvation, genes such as EPR1, CEPD1, CEPD2, and CEPDL2, which are known to participate in the response to N starvation, demonstrate significant changes in their expression within the vasculature of Arabidopsis plants [9,11,13]. This indicates an intricate and highly responsive system that plants employ to cope with N deficiency.

Moreover, similar responses are observed under phosphorus (P) deficiency. Six subsets of miRNA399 and the transcriptional activator AtMYB2 in *Arabidopsis thaliana* show augmented expression in the vascular tissue of cotyledons and roots [74,75]. This evidence further reinforces the hypothesis that plants have evolved to deal with nutrient scarcity by adjusting gene expression. Sahu et al., (2020) added another facet to this understanding by applying microscopic techniques to underscore the root apex as a crucial regulation center for plant phosphorus homeostasis [30]. This demonstrates that different plant tissues can play distinctive roles in managing nutrient distribution and uptake. The picture becomes more complex with the discovery of iron (Fe) deficiency-induced signaling behaviors. Specifically, the IRON MAN (IMA) signal, known for its role in long-distance communication, is shown to be predominantly expressed within the vasculature of *Arabidopsis thaliana* under iron-deficient conditions [62]. In the recent past, Nishida et al., (2017) have further added to this body of knowledge by deciphering the mechanistic details of the differential gene expression in *Arabidopsis thaliana* under different nutrient deficiencies [76]. Their work elucidates the specific pathways and regulatory networks involved, thus underlining the critical role of plant vasculature in nutrient homeostasis.

Collectively, these findings underscore the central role the leaf vasculature assumes in the long-distance signaling and transduction of key nutrients, including N, P, and Fe, in plants. These discoveries help us to better understand the complex and dynamic strategies that plants deploy for nutrient management in response to the changing environmental conditions they encounter. As such, these insights could have significant implications for optimizing crop yield and improving agricultural practices in nutrient-limited environments.

## Data Availability

Not applicable.

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
