# Peer review of "A Review: Systemic Signaling in the Regulation of Plant Responses to Low N, P and Fe"

_plants, 2023, doi:10.3390/plants12152765_

Round 1

Reviewer 1 Report

This ms could be a valuable contribution in the field of plant physiology, but I am not really able to be sure because the language is not adequate.     

An obvious person to help with the English is Dr Sandy Lang who has a background in phloem physiology and has many years' experience in helping authors with their language: https://rescript.co.nz

I would be happy to review an edited text.

Author Response

Dear Reviewer,

Thank you very much for giving us an opportunity to further revise our manuscript, and we greatly appreciate your for your positive and constructive comments on our manuscript entitled "A Review: Plant Phloem Systems Play a Major Role in Long-Distance Signaling Transduction of N, P, and Fe".

We have carefully studied the proposals and have made correction according to your comments.  To further improve the style and language of our manuscript, it has been revised by MDPI English Editing Center.

We appreciate your consideration of our manuscript.

Reviewer 2 Report

This is interesting and relevant review article manuscript. Authors should add Sahu et al. (2020) paper showing the highest levels of Pi in cells of the root apex transition zone in which phloem unloed of sucrose is accpmplished.

Sahu A et al (2020) Spatial Profiles of Phosphate in Roots Indicate Developmental Control of Uptake, Recycling, and Sequestration. Plant Physiology 184, 2064-2077

Author Response

Dear Reviewer,

Thank you very much for giving us an opportunity to further revise our manuscript, and we greatly appreciate your for your positive and constructive comments on our manuscript entitled "A Review: Plant Phloem Systems Play a Major Role in Long-Distance Signaling Transduction of N, P, and Fe".

We have carefully studied the proposals and have made correction according to your comments.  We really appreciate your suggestion about Dr. Sahu's work, and we made improvement in line 156 and line 283 to clarify Dr. Sahu's contribution on Pi signaling and responses. To further improve the style and language of our manuscript, it has been revised by MDPI English Editing Center.

We appreciate your consideration of our manuscript.

Reviewer 3 Report

Dear authors,

this review would be a nice story. But there are critical points which have to be solved.

1. The english grammar is over wide text parts very disastrous, especially the title, the abstract and the introduction. Also a lot of typing errors spread over the whole MS. Only some single chapters are well written e.g. 3.3.

2. More than half of the cited literature is missing in the reference list e.g. Balzergue et al., 2017 or Mora-Macias et al., 2017.....

3. Species names must be written in italic. Abbreviations must be introduced once when they are mentioned the first time.

Before I will review the content of the MS, the authors have to rewrite and correct the mentioned things.

Author Response

Dear Reviewer,

Thank you very much for giving us an opportunity to further revise our manuscript, and we greatly appreciate your for your positive and constructive comments on our manuscript entitled "A Review: Plant Phloem Systems Play a Major Role in Long-Distance Signaling Transduction of N, P, and Fe".

We are sorry for the mistakes we have made in last submission and we have carefully studied the proposals and have made correction according to your comments. We made correction on the english writing and almost rewrite the manuscript. To further improve the style and language of our manuscript, it has been revised by MDPI English Editing Center. In addition, all of the reference were checked to be consistent to the main body of our manuscript.

We appreciate your consideration of our manuscript.

Round 2

Reviewer 1 Report

It is strange that the authors claim that  We have carefully studied the proposals and have made correction according to your comments.”  I made no comment on V1. It needed editing to fix the English and be comprehensible.  The claim suggests that they are making comments extracted from a manual on how to respond, in order to appear polite. It’s not a good thing to do in western cultures.

Unfortunately, I find that the content, although interesting, is minimal and adds little to what is already published.  The latter works are  much more detailed.  To be accepted for publication, the authors need to show that their text really adds to existing literature and is more than a summary of some existing reviews.

The logic is problematic in many places.  That may be the due to  the editing of English (I hope not!).  The text proposes that some process “could” occur (eg  L40, 50, 67, 81 etc).   That means that the process is merely possible,  that it need not always occur. Obviously that is not useful:  the hypothesis must be  “falsifiable” (refer Popper).  If there is no response the hypothesis is still true!  

Title –  and elsewhere. Concerning nitrogen, the element (eg N) is not detected by the physiology, it is a compound, as the authors well know (probably nitrate or ammonia).   Although it is common to read of NPK in the horticulture literature, it’s best to be accurate.  Of course the term that is used for P,  phosphate, Pi, is OK, and Fe for iron.

L11 – 23 - this is ambiguous,  because  not only “ nutrient element deficiency”  causes  ” Plant signaling transduction”, it  can occur  in response to other things too. 

-            nutrient element deficiency in plant vascular tissue” needs an English fix – the deficiency is not in the plant vascular tissue!

L61-62 is illogical – it is our knowledge about triggering that is the sentence’s  topic, not the triggering itself.

L69 “ … a probability that our proposed model is closely related to the CEP-dependent pathway.”  could be more succinct -  “suggests  that our proposed model relies on the  CEP-dependent pathway” 

L79 “.. Ota et al., (2014) 78 raised a model which proposes that nitrate presence.” could be

 “Ota et al., (2014) proposed that nitrate …”

L179 – needs citation – what studies? NB -  no citations were needed at L39.

Author Response

Response to Reviewer 1:

It is strange that the authors claim that  “We have carefully studied the proposals and have made correction according to your comments.”  I made no comment on V1. It needed editing to fix the English and be comprehensible.  The claim suggests that they are making comments extracted from a manual on how to respond, in order to appear polite. It’s not a good thing to do in western cultures. 

Dear reviewer, thank you very much for processing our manuscript, your sincere comments are highly appreciated. We are sorry for this misunderstanding, in fact, the proposals we mentioned was the report form listed on the reviewer’ comments (shown as the five-pointed star symbols).

Unfortunately, I find that the content, although interesting, is minimal and adds little to what is already published.  The latter works are much more detailed.  To be accepted for publication, the authors need to show that their text really adds to existing literature and is more than a summary of some existing reviews.

Thank you for your carefully work in process of reviewing the manuscript. Your sincere comments are highly appreciated. We did our best to address these issues mentioned below. We have add some publications newly published to reflect the latest developments in this area.

The logic is problematic in many places.  That may be the due to  the editing of English (I hope not!).  The text proposes that some process “could” occur (eg  L40, 50, 67, 81 etc).   That means that the process is merely possible,  that it need not always occur. Obviously that is not useful:  the hypothesis must be  “falsifiable” (refer Popper).  If there is no response the hypothesis is still true!  

Thank you for your constructive suggestion, I checked the history of English Editing and found the word you mentioned was already exists before English Editing. The reason we used this word was due to our bad habituation of the expression, and we have already removed or replaced them into other appropriate words. For example, in the abstract: “Recent works have shown that the phloem (could be) is a central regulator in plant growth and development by transporting both essential nutritional and long-distance signaling molecules be-tween different parts of the plant’s tissues”.

Title –  and elsewhere. Concerning nitrogen, the element (eg N) is not detected by the physiology, it is a compound, as the authors well know (probably nitrate or ammonia).   Although it is common to read of NPK in the horticulture literature, it’s best to be accurate.  Of course the term that is used for P,  phosphate, Pi, is OK, and Fe for iron. 

Thank you for your suggestion. We have carefully checked the whole manuscript and corrected all this kind of mistakes. For example, the title was corrected to “A Review: Plant Phloem Systems Play a Major Role in Long-Distance Signaling Transduction of Nitrogen, Phosphorous, and Iron.” In addition, we have add the full name before we first use the abbreviation like N, P and K.

L11 – 23 - this is ambiguous,  because  not only “ nutrient element deficiency”  causes  ” Plant signaling transduction”, it  can occur  in response to other things too.  

-           “ nutrient element deficiency in plant vascular tissue” needs an English fix – the deficiency is not in the plant vascular tissue! 

Thank you for your sincere suggestion. We have corrected the former sentence into “Plant signaling transduction occurs in plant vascular tissue to cope with plants nutrient element deficiency”

L61-62 is illogical – it is our knowledge about triggering that is the sentence’s  topic, not the triggering itself. 

Thank you for your suggestion. We have rewritten this sentence to make it logical. “In the CK-dependent pathway, NO3− itself appears to be the triggering signal of sys-temic N-demand signaling, since NO3− supplied roots are the main providers of tZ.”

L69 “ … a probability that our proposed model is closely related to the CEP-dependent pathway.”  could be more succinct -  “suggests  that our proposed model relies on the  CEP-dependent pathway”  

Thank you for your kindly suggestion. We have rewritten this sentence to “The existence of biosynthesis of CEP in N-deprived roots suggests that our proposed model is closely correlated to the CEP-dependent pathway.”

L79 “.. Ota et al., (2014) 78 raised a model which proposes that nitrate presence.” could be

 “Ota et al., (2014) proposed that nitrate …”

Thanks again for your kindly suggestion. We have rewritten this sentence to “Ota et al., (2014) proposed that nitrate presence triggers tZ accumulation in roots, which is subsequently transported to the shoots.” And we have carefully checked the manuscript to avoid this kind of verbose expression.

L179 – needs citation – what studies? NB -  no citations were needed at L39.

We are sorry for our negligence. And we have rewritten this sentence into “Recent studies indicated that integration between local and systemic signaling to Pi starvation response exists in plants (Liang et al., 2014).” And the citations in Line 39 were removed. Moreover, the whole manuscript was carefully checked to make sure they all have appropriate citations.

Reviewer 3 Report

Dear Authors,

The content of the MS entitled „A review: Plant phloem systems play a major role in long-distance signalling transduction of N, P, and Fe” is very chaotic and not well organized. Moreover, you will find a lot of statements pasted against each other without any explanation and connection. Also, the content reflects in the current state not well the statement of the title. Of course, in some parts you mentioned the vascular system but not in the way which one would expect from the title. You may not experts in the vascular system, which is OK, but you must inform yourself accordingly. You disregard a lot of important literature (especially in the Fe chapter). The English grammar is much better now, but still there are many things which do not fit to the journal style (e.g. literature list: missing references, chaotic order, missing information; also, a lot of abbreviations are not introduced in the text).

Specific comments:

Abstract

Line 22 – systemic or systematic? Systemic is the transport from one plant part to another plant part; systematic is methodologic, persecute a system e.g. in nomenclature.

Line 25 – signal transduction network

Introduction

Need more general information about xylem and phloem (e.g. transport directions; morphological features; how can elements be transported and in which form; pH-dependency and membrane step in case of phloem)

Lines 39 and 42 - again, signal transduction

Line 43-46 – combine both sentences

Line 37-43 – literature is missing

2. Perception and response of N status

Chaotic, better order by the two types (CEP for chapter 2.1 and tz for chapter 2.2) and not by pathway and molecules

Line 51-53 – biomass transport? What did you mean? Biomass cannot be transported! Poitout et al., 2018 demonstrated a transport of NO-3

Line 55 – polar growth of lateral roots. Please explain!

Line 63 – in response to

2.1 Pathways involved in N perception

One can not well follow, because you describe two coexisting systemic signalling pathways and thereby jump back and forth between both. For reader it would be better if you sort.

Line 68 – Abbreviation (CEP) must be introduced

Line 94 – open end! NRT2.1 expression result in what? What is NRT2.1 (transporter gene) and what about NRT1.1 and NRT3.1?

Lien 96 – Arabidopsis thaliana must be italic

2.2 Molecules involved in response to N status

Line 110 – systemic or systematic?

Line 112-115 – I do not understand this sentence. May be a verb is missing?

Line 115-118 – Where is it transported and how did it enter the vascular system?

Line 120 – produced in roots, but in which cells?

Line 121 – What is moving?

Line 124 – N-rice? You mean N-rise!

Line 132-140 – must be explained in more detail! Why indicated CEPDL2 a signalling pathway which is coordinated by CEP-CEPR-CEPD pathway?

Line 138-140 - ? rephrase and explain. The reference is missing!

Line 135-157 – uptake and transport of what?

3. Perception and response of P status

The chapter is much better, but also need some improvement.

3.1 local response to P starvation

Line 180-182 – sentence makes no sense. Please rephrase.

Line 201 – open end. What is the consequence of malate exudation?

Line 204 – Abbreviation of PDR2 must be introduced.

Line 205-209 – RAM exhaustion by callose independent manner; need explanation!

3.2. systemic response of P starvation

Line 226 – PSi (Abbreviation need an introduction)

Line 243 – act as regulator or play regulative roles

3.3 integration between local and systemic signalling

A lot of Abbreviations are not introduced! Please check!

Line 255-256 – literature is missing

Line 257-264 – please specify this statement by telling reasons for the hypothesised statement

Line 265-270 – literature is missing

Line 265.279 – this chapter is not understandable! Please rephrase and include direct evidences and findings which support a potential cross-talk.

3.4 hormone regulation of P status

Chaotic, please sort the chapter, rephrase and explain.

Line 282 – I can not find information about ABA in this chapter.

Line 282, 313 and 314 – jasmonic acid instead of jasmine acid

Line 285 – Abbreviation need introduction (MAX1 and LBO)

Line 286 – closely related or involved in strigolactone biosynthesis? If it is only closely related one can only speculate…

Line 291 – what does it mean when Strigolactone is increased in root xylem?

Line 291-295 – explanation is missing

4. perception and response of Fe status

Very little information on Fe compared to P and N. Please improve, there is much more known in the literature. E.g. Tsukamota et al., 2009 direct transport of Fe; Lucena et al., 2019 signals involved in Fe and P deficiencies; Zhai et al., 2014 phloem-specific iron transporter; Khan et al., 2018 sensing of iron availability in leaf vasculature……..

Kobayashi et al., 2021 is not in the literature list

5. vasculature plays the crucial role in N, P and Fe signalling

The title of the MS mentioned the major role of the vascular system, why did you bring at the end such nice information about mRNA. It must be sorted to the single N, P and Fe chapter.

Line 384 – K?

Parts of the chapter must be at the beginning of the MS, because that are general information which will introduce the reader to the MS topic.

Author Response

Dear Authors,

The content of the MS entitled “ A review: Plant phloem systems play a major role in long-distance signalling transduction of N, P, and Fe” is very chaotic and not well organized. Moreover, you will find a lot of statements pasted against each other without any explanation and connection. Also, the content reflects in the current state not well the statement of the title. Of course, in some parts you mentioned the vascular system but not in the way which one would expect from the title. You may not experts in the vascular system, which is OK, but you must inform yourself accordingly. You disregard a lot of important literature (especially in the Fe chapter). The English grammar is much better now, but still there are many things which do not fit to the journal style (e.g. literature list: missing references, chaotic order, missing information; also, a lot of abbreviations are not introduced in the text). 

Dear reviewer, thank you very much for processing our manuscript, your sincere comments are highly appreciated. We did our best to address these issues mentioned below.

Specific comments:

Abstract

Line 22 – systemic or systematic? Systemic is the transport from one plant part to another plant part; systematic is methodologic, persecute a system e.g. in nomenclature.

Thanks for your suggestion, we have carefully checked the use of words in the manuscript and replace “systematic” into “systemic”. For example, “In this review, we assessed recent progress in the characterization of phloem as an integrated local and systematic systemic long-distance signal transduction and regulation center.”

Line 25 – signal transduction network

Thanks for your suggestion, and we have replace “signalling transduction” into “signal transduction” in the whole manuscript.

Introduction 

Need more general information about xylem and phloem (e.g. transport directions; morphological features; how can elements be transported and in which form; pH-dependency and membrane step in case of phloem)

Lines 39 and 42 - again, signal transduction

Thanks for your suggestion, and we have replaced “signalling transduction” into “signal transduction” in the whole manuscript.

Line 43-46 – combine both sentences

We have combined these two sentences to “Nitrogen, mainly acquired by plants in nitrate (NO3-) form and crucial for plant development and growth, can be deficient under two circumstances depending on the uniformity in the soil: uniformly low or unevenly low.”

Line 37-43 – literature is missing

We are sorry for this negligence. The literature has been added in the new submission.

  1. Perception and response of N status

Chaotic, better order by the two types (CEP for chapter 2.1 and tz for chapter 2.2) and not by pathway and molecules

Line 51-53 – biomass transport? What did you mean? Biomass cannot be transported! Poitout et al., 2018 demonstrated a transport of NO-3

We are sorry for our negligence. This sentence has been corrected to “Poitout’s work demonstrated that uniformly low N levels result in directional biomass nutritional elements transportation from the shoots to roots”

Line 55 – polar growth of lateral roots. Please explain!

This sentence has been rewritten to “Poitout's research elucidates how plants, facing uniformly low nitrogen conditions, adapt via a process called 'polar growth', where they directionally transport biomass nutritional elements from shoots to roots, facilitated by plant vascular systems, to enhance root growth and nitrogen absorption.”

Line 63 – in response to

Thank you for your kindly suggestion. We have corrected the “in responding to” into “in response to” in the whole manuscript.

2.1 Pathways involved in N perception

One can not well follow, because you describe two coexisting systemic signalling pathways and thereby jump back and forth between both. For reader it would be better if you sort.

Line 68 – Abbreviation (CEP) must be introduced

We are sorry for our negligence, the full name of CEP was stated in later sentences, now it was corrected in the resubmission. And we have carefully checked the manuscript to avoid this mistake.

Line 94 – open end! NRT2.1 expression result in what? What is NRT2.1 (transporter gene) and what about NRT1.1 and NRT3.1?

This sentence has been rewritten in the resubmission. “CEPR (C-TERMINALLY ENCODED PEPTIDE RECEPTOR) perceives peptides (CEP) moving from the roots to the shoots, initiating a signal cascade that amplifies the expression of the NRT2.1 gene, as well as the NRT3.1 and NRT1.1 genes, collectively enhancing nitrate uptake in plants.”

Lien 96 – Arabidopsis thaliana must be italic

We are sorry for this mistake, and we have corrected this mistake in the whole manuscript.

2.2 Molecules involved in response to N status

Line 110 – systemic or systematic?

We are sorry for this mistake, and we have corrected it into “systemic” in resubmission.

Line 112-115 – I do not understand this sentence. May be a verb is missing?

This sentence was rewritten in the resubmission to be more logical. “After reviewing recent molecular studies on nitrogen perception, we discovered that the root architecture of Arabidopsis thaliana is influenced by a long-distance signalling pathway as previously described”.

Line 115-118 – Where is it transported and how did it enter the vascular system?

This sentence was rewritten in the resubmission to be more informative. “When rootlets lack nitrogen, they secrete small peptides. These peptides, transported via the plant's vascular system, reach the shoots of Arabidopsis thaliana, where they interact with leucine-rich repeat receptor kinases (LRR-RKs), initiating a response to the nitrogen deprivation.”

Line 120 – produced in roots, but in which cells?

This sentence was rewritten to be more informative. “In response to nitrogen deficiency, Arabidopsis plants produce leucine-rich repeat receptor kinases (LRR-RKs) in specific root cells, which then travel to the shoots to adapt to fluctuating local nitrogen levels, leading to observable growth retardation.”

Line 121 – What is moving? 

This sentence was rewritten to be more informative. “In response to nitrogen deficiency, Arabidopsis plants produce leucine-rich repeat receptor kinases (LRR-RKs) in specific root cells, which then travel to the shoots to adapt to fluctuating local nitrogen levels, leading to observable growth retardation.”

Line 124 – N-rice? You mean N-rise!

We are sorry for this stupid mistake, and we have corrected in resubmission.

Line 132-140 – must be explained in more detail! Why indicated CEPDL2 a signalling pathway which is coordinated by CEP-CEPR-CEPD pathway?

We have rewritten this sentence to make it more informative. “Phloem-mobile CEPD-like 2 (CEPDL2) polypeptides, a vital category of plant-specific proteins, are integral to the CEP-CEPR-CEPD signaling pathway. Transported via the phloem, these polypeptides are secreted in response to conditions like nitrogen defi-ciency and induce specific developmental responses and adaptive strategies in plants. The integration of CEPDL2 polypeptides into the CEP-CEPR-CEPD pathway, a crucial communication system in plants, underlines their significance in nutrient signaling and growth regulation (Patterson et al., 2020). This points to a highly regulated, intercon-nected signaling system within plants that adaptively responds to nutrient environ-ment changes.”

Line 138-140 - ? rephrase and explain. The reference is missing!

We have rewritten this sentence to make it more informative as below and the reference was added.

Line 135-157 – uptake and transport of what?

This sentence was rewritten in the resubmission to be more informative. “Reciprocal grafting analysis demonstrated that CEPDL2 is upregulated by low nitrogen signals in leaf vasculature tissue and subsequently transferred to roots, promoting nitrogen uptake and transport between the root and shoot. Mutant analysis further confirmed the critical role of CEPDL2 in nitrate uptake, transport, and overall plant biomass accumulation.”

  1. Perception and response of P status

The chapter is much better, but also need some improvement.

3.1 local response to P starvation

Line 180-182 – sentence makes no sense. Please rephrase.

This sentence was rephrase in the resubmission. “The regulation of soil phosphorus (Pi) condition involves two major signaling pathways: local signaling and systemic signaling, which collectively coordinate the plant's response to the availability of phosphorus in the soil.”

Line 201 – open end. What is the consequence of malate exudation?

This sentence was rewritten in the resubmission to show the consequence of malate exudation. “Moreover, when phosphate deficiency occurs, there is an observed upregulation of the transcription factor SENSITIVE TO PROTON RHIZOTOXICITY 1 (STOP1). This upregulation of STOP1 leads to the activation of the MALATE TRANSPORTER 1 (ALMT1), which facilitates the exudation of malate from the roots (Balzergue et al., 2017). In response to phosphate deficiency, plants employ the adaptive strategy of malate exudation to cope with limited phosphorus availability. This exudation alters rhizosphere pH, improves solubility of insoluble phosphorus compounds, and facilitates phosphorus acquisition from the soil. Known as rhizosphere acidification, this process enhances phosphorus availability and uptake by plants under phosphate-deficient conditions (Ryan et al., 2010).”

Line 204 – Abbreviation of PDR2 must be introduced.

We are sorry for this mistake, and the full name of PDR2-LPR1 was added in the resubmission.

Line 205-209 – RAM exhaustion by callose independent manner; need explanation!

Thanks for your kindly suggestion. The explanation was added in the resubmission to be more logical and informative.

3.2. systemic response of P starvation

Line 226 – PSi (Abbreviation need an introduction)

Thanks for your kindly suggestion. The full name of PSi was added in the resubmission.

Line 243 – act as regulator or play regulative roles

We are sorry for this gramma mistake, and this sentence has been rephrase in the resubmission.

3.3 integration between local and systemic signalling

A lot of Abbreviations are not introduced! Please check!

We are sorry for this kind of stupid mistake, and we have carefully checked the full name of abbreviations in the resubmission to avoid this mistake.

Line 255-256 – literature is missing

We are sorry for this mistake. The citation was added in the resubmission.

Line 257-264 – please specify this statement by telling reasons for the hypothesised statement

Thank you for your constructive suggestion, and this sentence was rewrite as “Mutant studies have revealed that under conditions of phosphate (Pi) starvation, the expression of phosphate starvation response (PSR) genes is regulated both locally and systemically, leading to significant alterations in root system architecture. This suggests the existence of a potential cross-talk between the local and systemic signaling pathways, indicating intricate communication and coordination between these two regulatory mechanisms (Sanchez-Calderon et al., 2006; Mora-Macías et al., 2017).”

Line 265-270 – literature is missing

Citation was added in the resubmission.

Line 265.279 – this chapter is not understandable! Please rephrase and include direct evidences and findings which support a potential cross-talk.

This chapter was rephrase and added some new evidences about the potential cross-talk. “Regulation of ALMT1-mediated malate exudation in the root apical meristem under local Pi starvation conditions is closely associated with the translocation of sucrose from the shoots to facilitate malate biosynthesis in plant roots, suggesting the existence of a coordinated systemic and local signaling mechanism for improved Pi utilization efficiency under low Pi conditions (Zhang et al., 2021). This coordinated response is further supported by studies on mutant stop1 and almt1 plants, which demonstrated the suppression of local and systemic responses, particularly in the PHR1-direct targets, indicating a potential cross-talk between local and systemic signaling pathways in regulating Pi utilization in plant root tips (Tian et al., 2021). Moreover, Péret et al. (2011) provided evidence of an interaction between systemic phosphate signaling and local Pi sensing mechanisms in Arabidopsis roots. Additionally, the involvement of microRNAs and transcription factors as potential mediators of the cross-talk between local and systemic Pi signaling pathways has been elucidated in studies by Liu et al. (2015) and Chen et al. (2019).”

3.4 hormone regulation of P status

Chaotic, please sort the chapter, rephrase and explain.

Line 282 – I can not find information about ABA in this chapter. 

This paragraph was rewrite to be more informative in the resubmission. “Many hormone classes participate in plant Pi starvation responses, including auxin, ethylene, jasmine acid, gibberellin, strigolactone, and abscisic acid (Scheible and Rojas-Triana, 2015). ABA, in particular, plays a crucial role in regulating plant responses to Pi deficiency. It has been observed that ABA levels increase in response to Pi starvation signals, contributing to the modulation of various physiological and developmental processes (Liu et al., 2015). Transcriptomic studies have revealed that ABA-responsive genes, such as those involved in stomatal closure and stress responses, are upregulated in plants subjected to Pi deficiency (Feng et al., 2020; Huang et al., 2016).”

Line 282, 313 and 314 – jasmonic acid instead of jasmine acid

We are sorry for these mistakes, and all “jasmine acid” were replaced to jasmonic acid in resubmission.

Line 285 – Abbreviation need introduction (MAX1 and LBO)

The introduction of abbreviations were added in the resubmission.

Line 286 – closely related or involved in strigolactone biosynthesis? If it is only closely related one can only speculate…

Thank you for your suggestion, it should be “involved in”, and it was corrected in the resubmission.

Line 291 – what does it mean when Strigolactone is increased in root xylem?

This sentence was rewritten to be more informative. “Through grafting studies, it has been demonstrated that strigolactones, biosynthesized in roots, trigger a suppression of shoot branching. In instances of phosphorus (Pi) deficiency, there is an observed accumulation of strigolactones within the roots, suggesting a potential escalation of strigolactone levels in the xylem exudates of these Pi-deprived plants. This finding implicates an increase in strigolactone concentration within the root xylem under phosphorus-deficient conditions”

Line 291-295 – explanation is missing

This paragraph was rewritten in resubmission to be more informative. “In addition to other mechanisms, auxin redistribution has been proposed as a key regu-lator of root system architecture in response to phosphorus (Pi) deficiency, acting as a modulator of growth dynamics (Nacry et al., 2005). Experimental observations have shown that under conditions of phosphorus deficiency, plants exhibit enhanced sensi-tivity to auxin application in comparison to control conditions (CK). This enhanced sen-sitivity is characterized by suppressed growth of the primary root and a more pro-nounced promotion of lateral root formation, a shift in growth patterns that is likely a strategic adaptation to facilitate nutrient acquisition under conditions of nutrient stress (López-Bucio et al., 2002). This suggests a critical role of auxin in mediating root system architectural changes in response to Pi deprivation.”

  1. perception and response of Fe status

Very little information on Fe compared to P and N. Please improve, there is much more known in the literature. E.g. Tsukamota et al., 2009 direct transport of Fe; Lucena et al., 2019 signals involved in Fe and P deficiencies; Zhai et al., 2014 phloem-specific iron transporter; Khan et al., 2018 sensing of iron availability in leaf vasculature……..

Kobayashi et al., 2021 is not in the literature list

We have rewritten this paragraph to be more informative, and we also added the above citations to our resubmission. Rewritten paragraph was as below:

“In a pioneering study, Grillet et al. (2018) elucidated a short C-terminal amino acid sequence consensus motif known as IRON MAN (IMA). This motif was identified as a highly conserved peptide sequence prevalent among angiosperms and found to play a pivotal role in iron (Fe) uptake. They discovered that when these IMA motifs were overexpressed, a stimulation of Fe uptake in the roots was evident. This resulted in a marked accumulation of iron throughout the plant, subsequently bolstering the plant's tolerance to iron deficiency. Intriguingly, contrary to plants where IMAs were overex-pressed, ima mutants demonstrated contrasting phenotypes. As highlighted by Koba-yashi et al. (2021), this stark contrast underscored the integral role of IMAs in iron ho-meostasis and the repercussions of its absence or disruption.

Building upon the foundation of earlier translational profiling work by Mustroph et al. (2009), which identified that AtIMA is specifically expressed in plant phloem, Tsukamoto et al. (2009) deepened our understanding by providing insights into the mechanisms underlying direct iron transport. Subsequently, Zhai et al. (2014) unveiled the role of phloem-specific iron transporters. Their findings reaffirmed the critical role of the phloem in iron distribution within the plant, thereby situating the phloem as a central player in plant iron homeostasis.

Lucena et al. (2019) broadened this landscape of knowledge further by exploring the signals involved in Fe and phosphorus (P) deficiencies. Their research linked these deficiency signals with the roles of IMAs, providing a more comprehensive under-standing of the interconnected pathways involved in plant nutrient deficiency respons-es. By constructing a fusion of a promoter of IMA1 and the Enhanced Yellow Fluores-cent Protein (promIMA1::EYFP), Grillet et al. (2018) were able to visualize the protein in situ. They reported that fluorescence was primarily located in the phloem of the whole plants, thereby leading to the conclusion that IMA1 could serve as a mobile sig-nal within the plant phloem.

Recently, Khan et al. (2018) furthered our understanding of iron homeostasis in plants by elaborating on the intricate sensing mechanisms of iron availability within the leaf vasculature. In their latest work, Liang et al. (2021) proposed an intriguing the-ory that the movability of numerous IMAs constitutes a negative feedback loop on plants’ Fe acquisition. They found that Fe is activated within plants' phloem under low Fe conditions, thus highlighting a complex regulatory system governing iron uptake and distribution. This sophisticated network of iron regulation in plants underscores the intricacies of nutrient homeostasis, opening new avenues for understanding and im-proving plant nutrition.”

  1. vasculature plays the crucial role in N, P and Fe signalling 

The title of the MS mentioned the major role of the vascular system, why did you bring at the end such nice information about mRNA. It must be sorted to the single N, P and Fe chapter. 

Line 384 – K?

We are sorry for this mistake, and we have corrected it into Fe.

Parts of the chapter must be at the beginning of the MS, because that are general information which will introduce the reader to the MS topic.

Thank you for your constructive suggestion. We have add some information about the importance of “vasculature on long distance signaling” in the Introduction part. However, after carefully consideration on the importance of the last chapter “5. Vasculature Plays the Crucial Role in N, P, and Fe Signaling”, we still think this chapter should be located in the end as a conclusion for the MS right after introducing all of the information about N, P and Fe signaling. Thus, we just did some polishing work on this chapter to make it more logical and informative.

Round 3

Reviewer 1 Report

My apologies for this very slow response, especially as I have little major significance to add. The paper is basically fine, several edits are needed for best result to fix some logic, and some repetition that could be omitted.

L65 - the role of phloem to modulate root metabolism is not mentioned, despite that focus of the paper. So it should be

L115 - "perceived" is the word

L152 - Pi should be the subject of the sentence, not root tips. Eg "Pi is acquired by root tips conveyed to ... etc"

BUT root tip is actively dividing and expanding tissue. Is that really where uptake occurs? I suspect the uptake is behind the extending tips, but sensing occurs at the tip – as you discuss later..

L183 : to write "certain experimental conditions" is a distraction, the reader wonders what they are! So, specify, or, omit that detail.

L186 I suggest the role of vasculature be explicit as part of the pathway

L190 “concentration in plants” .. which part? Is it really the whole plant?? Maybe “Pi status” would be a better term, it is not a local property.

L216-219 seems to be a repeat of previous text

L254 it would help to describe exactly how grafting can be informative. (maybe at L359, and refer the reader forward to there).

L282 .. enriched? (Compared to what ...  ?sand,…) ,, by the plant?

L341 between mature leaves and sink organs” What is “bidirectional” about the flow? It is toward sink organs. The direction changes if a sink becomes a source, a developmental change.

Author Response

Response to Reviewer 2#:

My apologies for this very slow response, especially as I have little major significance to add. The paper is basically fine, several edits are needed for best result to fix some logic, and some repetition that could be omitted.

Response: Thank you for processing our manuscript. We have thoroughly reviewed the comments and made corrections in our resubmitted manuscript.

L65 - the role of phloem to modulate root metabolism is not mentioned, despite that focus of the paper. So it should be

Response: Thank you for your suggestion. According to our understanding to your comments, we have made modification on reducing occurrence of phloem modulating root metabolism in the resubmission.

L115 - "perceived" is the word           

Response: We are sorry for this mistake, we have revised “perceives” into “perceived” mistake in the resubmission version, and we have also go through the manuscript to avoid this kind of mistake.

L152 - Pi should be the subject of the sentence, not root tips. Eg "Pi is acquired by root tips conveyed to ... etc". BUT root tip is actively dividing and expanding tissue. Is that really where uptake occurs? I suspect the uptake is behind the extending tips, but sensing occurs at the tip – as you discuss later..

Response: Thank you for your suggestion, we have revised this sentence into “Pi, obtained by plant root tips, is transported to the shoots via the xylem and subsequently redistributed to sink tissues through the phloem.”.

L183 to write "certain experimental conditions" is a distraction, the reader wonders what they are! So, specify, or, omit that detail.

Response: Thank you for your suggestion, this sentence has been deleted in the resubmission version.

L186 I suggest the role of vasculature be explicit as part of the pathway

Response: Thank you for your suggestion. According to our understanding to your comments, we have revised this sentence of Line186 in the resubmitted manuscript.

L190 “concentration in plants” .. which part? Is it really the whole plant?? Maybe “Pi status” would be a better term, it is not a local property.

Response: Thank you for your suggestion, we have corrected the “concentration in plants” into “Pi status”.

L216-219 seems to be a repeat of previous text

Response: Thank you for your constructive suggestion, we have noticed this issue and this part was delete in resubmission.

L254 it would help to describe exactly how grafting can be informative. (maybe at L359, and refer the reader forward to there).

Response: Thank you for your suggestion, we have rewritten the related sentence in Line359 to emphasize the importance of grafting study.

L282 .. enriched? (Compared to what ...  ?sand,…) ,, by the plant?

Response: Thank you for your suggestion, and this sentence was rewritten in the resubmission.

L341 between mature leaves and sink organs” What is “bidirectional” about the flow? It is toward sink organs. The direction changes if a sink becomes a source, a developmental change.

Response: Thank you for your suggestion, and this sentence was rewritten in the resubmission as below “Anatomical analyses of plant vasculature have provided insights into the distinct transport functions of xylem and phloem, essential components of the plant's vascular system. The xylem primarily carries water, minerals, and signaling molecules from the roots to the shoots. In contrast, the phloem facilitates transport with-in the plant, transporting nutrients and signaling molecules from source organs such as leaves to sink organs like roots, young leaves, and above-ground tissues (Zhang et al., 2018; Turgeon et al., 2009).”

Reviewer 3 Report

Dear authors,

You worked very well on the MS. Now it's fine.

Author Response

Response to Reviewer 3#:

Dear authors, You worked very well on the MS. Now it's fine.

Response: Thank your patience on processing our manuscript.